# Biomarkers: The Key to Enhancing Deep Brain Stimulation Treatment for Psychiatric Conditions

**DOI:** 10.3390/brainsci14111065

**Published:** 2024-10-26

**Authors:** Guillermo J. Bazarra Castro, Vicente Casitas, Carlos Martínez Macho, Alejandra Madero Pohlen, Amelia Álvarez-Salas, Enrique Barbero Pablos, Jose A. Fernández-Alén, Cristina V. Torres Díaz

**Affiliations:** 1Department of Neurosurgery, University Hospital La Princesa, 28006 Madrid, Spain; guillebazarra@hotmail.com (G.J.B.C.);; 2Department of Neurosurgery, Hospital de la Santa Creu i Sant Pau, 08025 Barcelona, Spain

**Keywords:** deep brain stimulation (DBS), psychosurgery, depression, obsessive–compulsive disorder (OCT), biomarkers, diffusion tensor imaging (DTI)

## Abstract

Background: Deep brain stimulation (DBS) is currently a promising technique for psychiatric patients with severe and treatment-resistant symptoms. However, the results to date have been quite heterogeneous, and the indications for psychosurgery with DBS remain in an experimental phase. One of the major challenges limiting the advancement of DBS in psychiatric disorders is the lack of objective criteria for diagnosing certain conditions, which are often based more on clinical scales rather than measurable biological markers. Additionally, there is a limited capacity to objectively assess treatment outcomes. Methods: This overview examines the literature on the available biomarkers in psychosurgery in relation to DBS, as well as other relevant biomarkers in psychiatry with potential applicability for this treatment modality. Results: There are five types of biomarkers: clinical/behavioral, omic, neuroimaging, electrophysiological, and neurobiochemical. The information provided by each biomarker within these categories is highly variable and may be relevant for diagnosis, response prediction, target selection, program adjustment, etc. Conclusions: A better understanding of biomarkers and their applications would allow DBS in psychosurgery to advance on a more objective basis, guided by the information provided by them and within the context of precision psychiatry.

## 1. Introduction

### 1.1. Deep Brain Stimulation (DBS) and Psychosurgery

Psychosurgery refers to the neurosurgical treatment of severe mental illnesses that are resistant to conventional treatments, such as medication and psychotherapy. In society, about 20% of people are diagnosed with major depression, and 2% with obsessive–compulsive disorder (OCD). Of these patients, 10–20% will not improve with conventional treatments, which are based on subjective clinical diagnoses and “trial and error” treatments with psychotropic drugs and psychotherapy [1].

Psychosurgery, although controversial in the past, has experienced a resurgence thanks to technological advances and a better understanding of the neurobiology of mental illness. Approaches include lesion techniques and neuromodulatory therapies, most notably deep brain stimulation (DBS). DBS, initially approved in 2002 for the treatment of Parkinson’s, has been used for other disorders such as dystonia, epilepsy, OCD, pathological aggressiveness, anorexia, Gilles de la Tourette, schizophrenia, and depression, although many of these indications are still experimental [2].

Mental illnesses present unique challenges that complicate the use of DBS. Symptoms include thoughts and behaviors that are difficult to quantify, making diagnostic criteria such as DSM-V or ICD-11 not always effective. In addition, psychiatric illnesses often present comorbidities, such as 80–90% of patients with depression also presenting anxiety, which exacerbates refractoriness to conventional treatments [3]. Recent advances in neuroscience have led to a paradigm shift, understanding mental illnesses as alterations in neural networks rather than isolated dysfunctions of brain structures [4].

The concept of DBS has evolved from the stimulation of a specific nucleus to the modulation of complex neural networks, which has generated interest in connectome studies to correlate brain connectivity with patient clinical data and optimize treatments. However, identifying the optimal stimulation target remains a challenge, and the mechanisms of action of DBS in psychiatric patients are not yet fully understood [5]. A more personalized approach to DBS, based on the individual patient profile, might be the way forward.

Recent studies conducted by Neumann et al. [6] throw some light on how DBS exerts its effects, particularly via network modulation. The key mechanisms involved, as proposed by this group, are as follows:Modulation of Local Neural Circuits: DBS affects both local synaptic connections and distributed neural networks. In the subthalamic nucleus (STN), a common target in Parkinson’s disease treatment, DBS can activate neuronal fibers, leading to both local and distant effects. This activation can be orthodromic (downstream) or antidromic (upstream), generating activation of both afferent and efferent projections.Effects on Synaptic Plasticity: DBS can modify short-term synaptic plasticity, modulating synaptic transmission in a frequency-dependent manner. In structures with predominantly glutamatergic inputs, like the thalamus, stimulation induces postsynaptic depolarization, while in structures with GABAergic inputs, like the globus pallidus internus (GPi), stimulation results in hyperpolarization and neuronal inhibition.Suppression and Synchronization of Pathological Oscillations: At the mesoscale level, DBS reduces pathological oscillatory activity in the beta frequency range (13–35 Hz), which is prevalent in Parkinson’s disease. The suppression of these oscillations correlates with improvements in motor symptoms. Additionally, DBS can generate a new resonant activity, known as evoked resonant neural activity (ERNA), which is proposed as a marker for therapeutic effects.Modulation of Distributed Neural Networks: At the macroscale level, DBS influences broader brain network connectivity, such as the thalamocortical and cortico–subcortical networks, leading to effects in regions like the primary motor cortex (M1) and other movement-related areas. These modulations can alter both the coherence and directionality of oscillatory activity in these networks.

In addition to this, there are also other mechanisms related to the effects produced by DBS. Our group has recently conducted a bibliographic review on this topic [7]. We found that DBS may promote neurogenesis, as seen in animal models where increased expression of markers for immature neurons has been observed, potentially enhancing plasticity and cognitive functions like memory. Regarding neuroprotection, DBS has shown promise in slowing the progression of neurodegenerative diseases like Parkinson’s, particularly when applied early. This effect is linked to increased levels of neurotrophic factors and cytokines, creating a favorable environment for neuronal survival. DBS also leads to changes in neurotransmitters, especially dopamine, which plays a crucial role in its therapeutic effects on movement disorders. Additionally, the stimulation affects serotonin and norepinephrine, contributing to mood regulation and motor control. The extracellular microenvironment is also altered by DBS. Electrical stimulation activates astrocytes, which release neurotransmitters like glutamate, and modifies the extracellular matrix by releasing proteins such as IGFBP3 and PAPPA. These changes help sustain DBS’s therapeutic effects even after stimulation has ceased. This suggests that DBS not only corrects abnormal neuronal firing but also promotes neuroprotection, regeneration, and significant neurotransmitter regulation.

### 1.2. Precision Psychiatry

Precision medicine relies on managing large amounts of data available from patients, including their genetics, environment, and lifestyle, to tailor treatments based on their individual variability, ensuring more accurate and effective healthcare solutions. This approach has transformed disciplines such as oncology, where treatments are decided based on genetic and molecular biomarkers [8]. However, psychiatry has not advanced at the same pace due to the complexity of mental illness, where symptoms vary widely between patients with the same diagnosis [9].

The precision approach in psychiatry seeks to identify biomarkers for more personalized diagnosis and treatment. “Omics” techniques, such as genomics and proteomics, together with neuroimaging and clinical data, could contribute to the development of biosignatures for more accurate classification of mental illnesses [10]. Biomarkers often correlate with genetic factors that can lead to neurotransmitter imbalances, ultimately reflecting in the neuroimaging and clinical symptoms observed. For example, variations in the serotonin transporter gene (5-HTTLPR) have been linked to increased risks of anxiety and depression [11]. Such genetic differences can influence brain processing, as neuroimaging studies have shown altered amygdala activity. This interaction and modification of these factors through neuromodulation might allow for predicting responses to treatments such as deep brain stimulation (DBS) and optimizing stimulation targets. Although precision psychiatry is still in its early stages, it holds promise for improved prognostic predictions and treatment optimization.

### 1.3. Biomarkers in Psychosurgery

In recent decades, there has been an increased interest in biomarkers in mental illness, raising the possibility that diagnosis based on clinical criteria, such as the DSM, lacks biological validity [12]. The use of biomarkers in diagnosis and treatment could revolutionize psychiatry, although their identification is not easy due to the high heterogeneity of results in the literature. Although biomarkers of response to antidepressants and psychotherapy have been proposed outside the field of DBS, in psychosurgery, the identification of these is more complicated due to the smaller number of patients treated and the lesser experience in the application of these modern techniques [13].

The ideal process to create a biomarker in psychosurgery with DBS includes four phases: identifying a phenotype that can be measured and modified with DBS, validating the biomarker intra- and interindividually, associating it with the response to DBS through clinical trials, and conducting multicenter studies to confirm its usefulness. Biomarkers can provide diagnostic, response, monitoring, prognostic, and safety information [5]. Biomarker-based precision psychiatry could transform the use of DBS in psychosurgery, allowing for more personalized neuromodulation tailored to the patient’s profile.

### 1.4. Deep Brain Stimulation Based on Precision Psychiatry

DBS treatment in psychiatric disorders has shown a large variability in results, influenced by several factors such as the selection of the stimulation target and the programming of the electrodes. The identification of biomarkers based on the patient’s phenotype would allow treating individuals in a more personalized way, improving the prediction of the response to treatment. The development of connectome atlases, which correlate brain connectivity with the patient’s symptoms, could help define a more precise neuromodulation [14]. Advancing in this concept, symptom networks, such as those defined in anxiety, depression, and OCD, could guide the selection of stimulation targets more appropriate for each patient.

## 2. Objectives

In this work, we aim to review the existing literature on biomarkers related to deep brain stimulation in the treatment of psychiatric disorders, as well as other biomarkers already described in these diseases, which could have a possible future application in psychosurgery using this therapeutic method.

### Materials and Methods

In this study, a literature review on biomarkers in psychosurgery with deep brain stimulation (DBS) was carried out.

The articles were obtained through searches in PubMed, Embase, and Google Scholar. In addition, the search was expanded by reviewing the references cited in these articles. The time limit for the search was set on 12 September 2024.

Only articles in English were included. In the search process, articles were selected based on keywords: Deep brain stimulation, DBS, biomarkers, connectivity, tractography, functional RMI, fRMI, disorders, psychiatric depression, OCD, Tourette syndrome, electrophysiological, electroencephalography, neurochemical, personalized, precision, closed-loops.

After the search, articles referring to biomarkers in psychosurgery with deep brain stimulation were included. Furthermore, the review was extended to those articles that showed relevant information about biomarkers in psychiatric diseases and that, although they were not described in relation to DBS in psychosurgery, were considered interesting to include as potential biomarkers for study and future application in this field.

Articles referring to DBS in movement disorders, non-invasive neuromodulation, or psychosurgery with ablative techniques, which did not include relevant information about DBS in psychosurgery, were excluded based on the title and abstract.

## 3. Results

Biomarkers related to DBS treatment can be classified into the following categories: clinical or behavioral biomarkers, peripheral biomarkers, neuroimaging biomarkers, electrophysiological biomarkers, and neurobiochemical biomarkers. Below, we will review biomarkers for DBS in psychosurgery within each of these categories.

### 3.1. Clinical/Behavioral Biomarkers

In the absence of objective biomarkers, subjective rating scales remain the standard in psychiatry to measure disease severity and monitor treatment [1].

These scales, however, have limitations for assessing outcomes in neurosurgery. An example is major depression where the most common scales (HAM-D and MADRS) are not ideal in psychosurgery for two reasons:Patients undergoing neuromodulation are refractory to conventional treatment, and these scales were not designed to assess resistant depression or to monitor long-term outcomes [1].They can be imprecise, influenced by external factors (such as recent stressful events), leading to variations in outcomes that do not reflect actual clinical effectiveness [15].

In addition, the scales do not break down key aspects of depression (anhedonia, emotional dysregulation), which may be more relevant to clinical response in DBS. This might have contributed to the failure of clinical trials in psychosurgery due to the lack of adequate clinical biomarkers, rather than treatment inefficacy [1].

Regarding OCD, Rios-Lagos and colleagues (among whom the last author is included) have recently conducted a study in which we sought to investigate the hypothesis about processing speed in patients with treatment-resistant OCD and to clarify to what extent slowness is related to psychopathological symptoms. A clinical and neuropsychological examination was performed on 39 patients with resistant OCD, candidates for neurological surgery, and 39 matched healthy individuals. Principal component analysis revealed a three-component structure in the neuropsychological battery used, which included processing speed, working memory, and conflict monitoring. Comparisons between groups showed that OCD patients performed significantly worse than healthy individuals on speed measures, but no differences were found on executive tests not influenced by time. Correlation analyses revealed a lack of association between neuropsychological and clinical measures. The results suggest that patients with treatment-resistant OCD present a primary deficit in information processing speed, independent of clinical symptoms [16].

Finally, given the advance in knowledge of neural networks, it is suggested that clinical scales be adapted to measure changes in modulated networks, rather than focusing only on general pathology [1].

### 3.2. Peripheral Biomarkers

No peripheral biomarkers have been identified in psychiatric patients treated with DBS, but there are studies suggesting their possible future use in psychosurgery, which we will summarize below as follows:

#### 3.2.1. Serum Biomarkers

Several biomarkers can be measured in blood (plasma), although many have limited evidence and low specificity. Most of them have been used with a diagnostic approach, and their level changes related to treatment response or resistance are still unknown. Some of the most relevant biomarkers are broken down below for schizophrenia, depression, and OCD [13,17,18]:Increased in schizophrenia: arachidonic acid, antigliadin IgA, anti-NMDAR antibodies, malondialdehyde, and sIL-2 receptor.Decreased in schizophrenia: adiponectin, vitamin B6, NGF, and TNF-alpha.Increased in depression: C-reactive protein, FGF-2, glutamate, IL-6, IGF-1, lipid peroxidation marker, and sIL-2 receptor.Decreased in depression: BDNF, KYNA/3HK, KINA/QUIN, and KINACID.Increased in OCD: cortisol, glutathione persoxidase, superoxide dismutase, 8-hydroxydeoxyguanosine, and malondialdehyde.Decreased in OCD: vitamin C and vitamin E.

#### 3.2.2. Genomic Biomarkers

Genetics plays an important role in mental illness, but its clinical applicability is limited by polygenic complexity. GWAS studies have generated polygenic risk scores (PRS) to predict the evolution of diseases such as schizophrenia or the response to lithium in bipolar disorder. Although promising, it is not yet clinically useful but could be useful in the future to identify responses to DBS [19].

#### 3.2.3. Transcriptomic Biomarkers

In depression, changes in the expression of genes related to a better response to antidepressants (MMO28 and KXD1 genes) have been identified [20]. MicroRNAs (miRNA-146a-5p, miR-146b-5p, miR24-3p, and miR-425-3p) have also been linked to response to antidepressant treatment, suicide risk, and substance abuse [21].

#### 3.2.4. Proteomic Biomarkers

Proteins related to neuronal transmission and other processes have been proposed as biomarkers in schizophrenia [22]. Zinc finger protein 729 was found to be decreased in patients with schizophrenia compared to the controls [21]. GMF-beta, BDNF, and RAB3GAP1 were also found to be decreased in this pathology [23]. In depression, acetyl-L-carnitine is a marker of severity and resistance to treatment [24]. In anorexia nervosa, neurofilament light chains (NF-L) have been associated with neuronal damage [25].

#### 3.2.5. Metabolomic Biomarkers

Small molecules of cellular metabolism are also considered potential biomarkers. In depression, certain levels of phosphatidylcholine C38:1 (absence of response) and hydroxylated sphingomyelin (increased response) in plasma have been associated with the prognosis and recovery of symptoms after treatment [26].

#### 3.2.6. Epigenetic Biomarkers

Epigenetic modifications, such as gene hypermethylation, have been linked to depression, schizophrenia, and post-traumatic stress. In addition, methylation of genes such as BDNF or FKBP5 could be a predictor of response to antidepressants [27].

### 3.3. Biomarkers in Neuroimaging

The use of imaging techniques, such as functional MRI and tractography, has been key to conceptualizing DBS as a modulation of neural networks, rather than a specific nucleus. These techniques allow the analysis of both the local impact of stimulation and that of the anatomically and functionally connected regions [28].

Currently, neuroimaging plays a crucial role in presurgical screening, target selection, neurosurgical planning, electrode localization, therapeutic evaluation, and analysis of clinical correlations [29].

#### 3.3.1. Obsessive–Compulsive Disorder (OCD)

In OCD, alterations in volume, metabolism, and blood flow are observed in structures such as the orbitofrontal cortex, anterior cingulate cortex, caudate, amygdala, and prefrontal cortex, all members of the cortico–striato–thalamic–cortical network [30]. A larger volume of the nucleus accumbens (NAcc) has been proposed as a biomarker of response to DBS [31], and reduced connectivity between this nucleus and the medial/lateral prefrontal cortex has been associated with clinical improvement [32].

Connectivity with areas such as the middle frontal gyrus and the frontothalamic network correlates with a good response to DBS. Furthermore, stimulation of the ventral anterior limb of the internal capsule (vALIC) improves anxiety and mood symptoms [33]. On the other hand, stimulation in targets such as the VC/VS activates several structures of the cortico–striato–pallidum–thalamus–cortical network, while connectivity with the hypothalamus can produce adverse effects, such as weight gain [34]. Given these findings, some authors advocate grouping the NAcc, VC/VS, and ALIC targets as a “striatal region” given the structural and functional proximity [3].

The medial forebrain fasciculus (slMFB) has shown better results when stimulating areas close to the slMFB rather than the anterior thalamic radiations (ATRs) [35]. In the subthalamic nucleus (STN), stimulation reduces metabolism in areas such as the cingulate and medial frontal cortex, although it may also increase impulsivity [36]. Measurements of the metabolism in those targeted areas might be potential biomarkers for the selection of patients for DBS.

Simultaneous stimulation of amSTN and VC/VS has shown improvement without added effects from their combined stimulation. DBS in the amSTN improved cognitive flexibility, while in VC/VS, an improvement in mood was observed [37].

A study on the structural connectivity of the bed nucleus of the stria terminalis (BNST) in OCD showed a relationship between connectivity in this area and a reduction in the Y-BOCS scale when stimulating subcortical pathways connected to the amygdala, hippocampus, and stria terminalis, as well as cortical areas such as the prefrontal cortex, parahippocampus, and extrastriate visual cortex [34].

#### 3.3.2. Major Depression

Depression involves the dysfunction of several limbic networks connected to the DMN (default mode network) [38]. An increase in metabolism in the subcallosal gyrus has been proposed as a biomarker of response in depression, especially in the stimulation of this region (Cg25) [39]. Reduced activity in the dorsal anterior cingulate cortex, posterior cingulate cortex, and precuneus predicts clinical improvement [40].

The medial forebrain fasciculus is another target in depression, showing correlation between frontopolar/orbitofrontal volumes and clinical response, which could personalize treatment [41].

In patients with stimulation in the NAcc, an alteration in the frontostriatal network was observed, which is also involved in depression. Hyperactivation was identified in the dorsolateral prefrontal cortex, cingulate cortex, and amygdala, and hypoactivation in the ventromedial and ventrolateral prefrontal cortex, dorsal caudate, and thalamus [42].

Other target areas in DBS to treat depression include the ALIC and the VC/VS complex, which have shown improvement in depressive symptoms in patients with OCD. Due to their connections with the bed nucleus of the stria terminalis and the NAcc, these areas could influence stress regulation and reward and motivation management [43].

Complete remission of depressive symptoms has also been described in one patient after stimulation in the lateral habenula (LHb), in which hyperactivity is described in patients with depression [44].

Stimulation of the inferior thalamic peduncle has also shown a reduction in depressive symptoms, although in few patients [45].

#### 3.3.3. Tourette Syndrome

Stimulation in the internal globus pallidus (GPi) improves tics based on its connectivity with limbic and associative networks [46]. Stimulation in the centromedian–parafascicular nucleus (CM-Pf) reduces motor and vocal tics by suppressing motor and insular hyperactivation [47].

The internal thalamic–centromedian–ventro–oral nucleus (CM-VOI) showed better results in patients with activation of fibers projected to premotor areas, especially Pre-SMA. Wider stimulations in Pre-SMA, SMA, and M1 were associated with worse responses [48].

In a study using CM/VC/VS targets, patients who improved clinically showed increased connectivity with the precentral gyrus, while dizziness was related to cerebello-rubral fibers, paresthesias to thalamic and insular connections, and depression to fibers connecting the thalamus and amygdala [49].

#### 3.3.4. Anorexia Nervosa

DBS over the NAcc has shown a reduction in frontal, limbic, and insular hypermetabolism, while stimulation in Cg25 improves affective and body perception symptoms [50].

#### 3.3.5. Addiction, Schizophrenia, and Post-Traumatic Stress Disorder

In addiction, the NAcc is the main target due to its involvement in the dopaminergic reward system [51]. In PTSD, the basolateral amygdala has been proposed as a target [52], while in schizophrenia, the NAcc and Cg25 have been explored without conclusive biomarkers to date.

### 3.4. Electrophysiological Markers

EEG is a useful tool as a biomarker in psychosurgery, providing preoperative, intraoperative, and postoperative information, especially in closed-loop systems [52].

#### 3.4.1. Preoperative Markers

In patients with OCD, the surface EEG shows frontal asymmetry in alpha and beta ranges, increased ERN (error-related negativity), and REM sleep disturbances [53]. An elevated ERN correlates with limited success of the intervention. It has also been observed that a decrease in the beta band in the anterior cingulate cortex is a favorable indicator of response to treatment [54]. Decreases in the beta band in the anterior cingulate cortex and medial frontal gyrus have also been described for this entity as biomarkers of a favorable response to medical treatment [55].

In depression, the EEG shows hyperactivity of the anterior cingulate cortex, detected by theta activity in areas such as Fz and FCz, using techniques such as LORETA for greater precision [56].

In schizophrenia, alterations in the synchronization of neural oscillations, particularly in the beta and gamma frequencies, have been identified as key factors in the disorder’s pathophysiology. These oscillations are fundamental for coordinating neuronal responses across the cortex, impacting cognitive and perceptual functions. Studies have shown that patients with schizophrenia exhibit decreased synchronization and amplitude of high-frequency oscillations, contributing to the cognitive deficits characteristic of the disorder [57,58]. These abnormalities are attributed to deficits in GABAergic interneurons and corticocortical connections, which hinder the generation of coherent oscillatory activity. These patterns of cortical disconnectivity could be potential targets for DBS interventions, as this technique may restore altered neuronal synchronization and improve the affected cognitive functions [57]. These oscillatory deficits not only impact sensory and perceptual responses but also more complex functions such as working memory and perceptual organization. This suggests that therapies aimed at restoring neural oscillation coherence could provide a promising approach to improving cognitive symptoms in schizophrenia [58].

Additionally, alterations in oscillatory profiles and connectivity patterns in schizophrenia suggest a functional disorganization in the synchronization of brain networks, which negatively impacts cognitive functions. These disruptions involve both the modulation of gamma oscillations (~40 Hz), essential for cognitive processing, and the coordination between gamma and theta frequencies, which is crucial for integrating information across different temporal scales. Schizophrenia patients exhibit reduced gamma phase coherence and increased theta wave amplitude, though they maintain intact hierarchical modulation between the two frequencies [59]. Abnormalities in gamma oscillations are associated with deficits in perception, attention, and memory, suggesting that schizophrenia affects the ability of neural networks to synchronize oscillatory activity, which is critical for coordination between cortical areas [60]. In this context, Friston’s “dysconnection hypothesis” emphasizes that schizophrenia is a disorder of functional connectivity, affecting synaptic modulation, particularly mediated by NMDA receptors and dopaminergic systems, which contributes to false inferences and cognitive deficits [60]. These connectivity disturbances could be key targets for therapeutic interventions, such as deep brain stimulation (DBS), which may restore balance in neuronal synchronization by influencing brain oscillation modulation and enhancing synaptic plasticity. DBS, by modulating dysfunctional circuits, offers the potential to correct abnormal processing in neural networks, optimizing connectivity between cortical and subcortical regions, with potential benefits for cognition and reducing psychotic symptoms [59,60].

#### 3.4.2. Intraoperative Markers

In patients with OCD, intraoperative recordings reveal abnormal activity in the caudate nucleus and aberrant local field potentials in frontal electrodes [32]. In subjects with depression and DBS in the subgenual gyrus, autonomic phenomena such as tachycardia and increased skin conductance have been observed, correlated with a good response to stimulation [61].

#### 3.4.3. Postoperative Markers

In OCD, NAcc DBS shows reduced frontal delta oscillation, correlated with the severity of obsessions and compulsions [32]. Normalization of P300 amplitude is also observed in patients with a good response to serotonin reuptake inhibitors, which is associated with better results in working memory and attention [62]. Another study conducted by Schwabe et al. [63] showed that DBS significantly reduces oscillations in the theta band (4–8 Hz) in the BNST/ALIC and frontal cortex, suggesting that enhanced synchronization in this band is related to OCD symptom severity. While theta activity decreases, DBS increases oscillations in other frequency bands (alpha, beta, gamma). Notably, beta band activity is associated with anxiety, which aligns with patient reports of increased anxiety with higher stimulation intensities.

DBS targeting the BNST/ALIC has emerged as a promising therapeutic option for patients with trOCD. This study shows that DBS modulates neuronal activity in key brain regions involved in OCD, supporting the hypothesis that changes in oscillatory synchronization could serve as biomarkers to monitor treatment efficacy.

In refractory depression, action potentials in the bed nucleus of the stria terminalis and subgenual alpha activity allow differentiation between OCD and treatment-resistant depression [64]. Alpha activity in the subgenual area has been found to correlate with the severity of depression, while beta desynchronization has an inverse correlation [65].

In patients with depression treated with DBS in the subgenual gyrus, changes in frontal theta and parietal alpha activity have been shown to be predictors of depression severity before and after treatment [30]. In addition, biomarkers such as low theta energy in the subgenual area and high parietotemporal alpha energy have been identified, although some results have not been replicated in other studies [66].

In Tourette syndrome, an increase in low frequency and alpha range has been observed in the thalamus and GPi, which serves as a biomarker of disease severity [67]. In another study, an increase in the gamma band in the median center of the thalamus has been shown after symptom improvement [68]. In patients with alcohol addiction treated with DBS in the NAcc, the increase in ERN has been shown to correlate with clinical improvement [69].

### 3.5. Neurobiochemical Markers

Neurobiochemical biomarkers measure changes in extracellular neurotransmitters, providing key information about the disease status thanks to their high temporal resolution.

In OCD, DBS in the NAcc modulates the cortico–striato–thalamus–cortical network and regulates neurotransmitter levels. This includes an increase in dopamine, serotonin, and norepinephrine in the prefrontal cortex, an increase in dopamine in the striatum, and an increase in dopamine and GABA with a decrease in glutamate in the NAcc [70]. These changes are clinically relevant. Techniques such as voltammetry and biosensors allow the measurement of dopamine, serotonin, and glutamate with high precision, making them good candidates for closed-loop systems in OCD [70].

In depression, serotonin and dopamine play a crucial role in the pathophysiology and treatment, with alterations observed in the striatum and hippocampus, making them potential neurochemical monitoring regions [71,72].

In Tourette syndrome, there are abnormal concentrations of dopamine, GABA, and glutamate, with dopamine being the most promising biomarker. This is due to the positive response to dopaminergic antagonists and the observation of hypoactivity in the striatum, suggesting that measuring dopamine in this region is useful for future closed-loop systems in Tourette [70].

### 3.6. Closed-Loop Systems

Closed-loop systems in neurostimulation represent an innovative approach by continuously adjusting stimulation parameters in real time, based on feedback from specific biomarkers that reflect the patient’s clinical condition. This contrasts with traditional open-loop systems, where parameters such as amplitude, frequency, and duration are pre-programmed and manually adjusted according to the clinician’s evaluation of the patient’s response. These manual adjustments, however, can be insufficient due to the dynamic and fluctuating nature of neurological diseases, making open-loop systems less adaptable and less effective in real-time treatment [70].

The major advantage of closed-loop systems lies in their ability to use data from biomarkers—such as electrophysiological signals and neurochemical measurements—to automatically fine-tune stimulation parameters based on the patient’s neurophysiological state. This allows for a more personalized treatment, as the system can adjust its behavior depending on the individual patient’s needs at any given moment. Biomarkers such as action potentials, neurotransmitter levels (including dopamine, serotonin, and glutamate), and neural oscillations provide continuous and precise feedback that enhances the effectiveness of treatment [70].

For closed-loop systems to function optimally, three critical characteristics of biomarkers are required: high specificity, a strong signal-to-noise ratio, and high temporal resolution. High specificity ensures that the biomarker signals reflect the patient’s clinical state accurately without being confused by irrelevant data. A strong signal-to-noise ratio is essential to filter out interference or “noise”, which can distort the signals and lead to inaccurate adjustments. High temporal resolution enables the system to capture rapid neurophysiological changes, ensuring timely and precise modifications to the stimulation parameters [70].

The difficulty of replicating results and the lack of uniform criteria for biomarkers in psychiatric diseases complicate the self-programming of closed-loop systems. Currently, these systems are contributing experimentally to the understanding of the circuits and electrical activity involved in mental disorders [73].

An example is the use of field potentials in the CM-Pf in patients with Tourette syndrome, where an increase in low-frequency and alpha-range activity was found [67]. In another study, the authors identified an electrophysiological biomarker for depression in a patient who received bilateral DBS in the anterior limb of the internal capsule (ALIC). They used the left electrode to administer stimulation and the right electrode to record local field potentials, alternating the stimulation on and off. They discovered that stimulation in the left ALIC generated a broadband power increase in the right bed nucleus of the stria terminalis (BNST), including significant increases in both high and low gamma power, accompanied by an overall improvement in symptoms. Additionally, they found a significant inverse correlation between high and low gamma power and depression severity, as measured by the Visual Analog Scale (VAS) for depression. These findings suggest that personalized gamma activity could serve as a biomarker for depression, potentially paving the way for closed-loop DBS systems that adjust treatment in real time based on the patient’s symptoms, enhancing the efficacy of the therapy [74].

Another interesting approach was presented by Scangos et al., who implanted electrodes in several brain areas of a patient with severe depression. They found that gamma activity in the amygdala indicated the severity of the disease, while the VC/VS was the area with the greatest symptomatic improvement. A connection was identified between the VC/VS and the amygdala, suggesting that the recording and stimulation areas may be different [4].

In the future, it is hoped to move towards more precise and effective closed-loop systems, based on reliable biomarkers for the treatment of psychiatric diseases.

## 4. Discussion

Psychiatric illnesses represent a major cause of chronic morbidity, with a major socioeconomic impact. Although conventional treatment includes psychotherapy and medications, up to 30% of patients are refractory to these therapies. For these severe cases, alternative treatments such as psychosurgery, including deep brain stimulation (DBS), are considered. Despite technological advances, DBS in psychiatry is still in the early stages of development [75].

One of the greatest challenges in this field is the heterogeneity of psychiatric disorders, since clinical diagnoses are based on changing criteria (such as those of the DSM) and some symptoms occur in multiple diseases. This situation makes both diagnosis and treatment difficult [76].

Therefore, the goal is to move towards precision psychiatry, where biomarkers are identified that allow the treatment of each patient to be personalized according to their individual profile or “biosignature”. These biomarkers can be classified into categories based on the information they provide as follows:Diagnostic biomarkers: translation of a clinical profile of psychiatric symptoms into a biosignature of objective markers.Prediction biomarker: individual estimation of the effectiveness of DBS compared to other therapies and assessment of the pretreatment effectiveness of each of the targets.Prognostic biomarkers: prediction of patient outcome after DBS treatment.Safety biomarkers: identification of patient characteristics that are not favorable for DBS.Susceptibility/risk biomarkers: assess the probability of complications derived from DBS.Biomarkers of response: after the insertion of DBS and the initial programming, evaluation of the improvement/refractoriness of the treatment.Monitoring biomarkers: monitoring the patient with temporal fluctuations in response and making programming adjustments accordingly.

While this model is still hypothetical, it presents a promising future for personalized therapies driven by biomarkers and advanced technology. Taking into account the proposed scheme, we can develop a possible workflow of precision psychiatry applied to DBS.

The process begins with a comprehensive clinical evaluation of a patient with a severe psychiatric condition. Initially, a psychiatrist analyzes the patient’s clinical and behavioral profile, using scales or symptom clusters designed to correlate with specific biomarkers. Following this, a panomic study is conducted, which includes genomic, transcriptomic, proteomic, metabolomic, and epigenetic analyses, alongside neuroimaging and non-invasive electrophysiology tests. The integration of these studies results in the creation of a personalized “biosignature” for the patient, which helps identify the dysfunctional neural networks underlying their condition.

This biosignature introduces a novel diagnostic approach, focusing on the dysfunction of specific neural circuits rather than relying on standard clinical criteria like the DSM-V. This shift in diagnostic methodology has the potential to revolutionize psychiatry, as it could lead to the modification or elimination of current diagnoses, and even the creation of new, more precise diagnostic categories. For instance, patients presenting with symptoms of multiple disorders, such as psychotic depression or co-occurring conditions like obsessive–compulsive disorder (OCD) and depression, could receive more objective and personalized treatments targeting the dominant dysfunctional network in their case.

Once the biosignature is established and a hypothesis regarding the affected neural network is formulated, the next step is to predict the efficacy of various treatments. This involves evaluating the likelihood of success for different therapeutic options, including medications, psychotherapy, or neurosurgical techniques, and identifying patients who are unlikely to respond to conventional therapies and for whom DBS may be the best option. Biomarkers are also used to pinpoint the most appropriate brain targets for therapeutic intervention.

In conditions like treatment-resistant depression, there is already enough neurobiological knowledge to understand that this is a heterogeneous entity, with different neural networks involved in various clinical profiles. Proposed networks include the default mode network, salience network, negative and positive affective networks (related to reward), the attention circuit, and the cognitive control circuit. Instead of treating depression as a uniform condition, this approach seeks to directly target the specific neural network dysfunction that is most relevant to each patient.

Once the altered neural network is identified, surgical intervention proceeds with the insertion of electrodes for deep brain stimulation. The aim is to direct DBS to the most appropriate brain region based on the patient’s biosignature. To ensure precise electrode placement, imaging biomarkers, particularly tractography, are utilized to select not just an anatomical structure but a patient-specific tractographic region. During and after surgery, additional data are collected through recording systems that provide complementary information on the patient’s progress and treatment effectiveness.

In addition to preoperative studies, treatment adjustments are made postoperatively and throughout long-term follow-up, using biomarkers to monitor the therapeutic response and disease progression. This continuous follow-up, with periodic studies of omics biomarkers and neuroimaging, allows for treatment modifications according to any fluctuations caused by changes in the disease state or environmental factors.

Furthermore, before initiating treatment, safety biomarkers are evaluated to determine if the patient is a good candidate for DBS. Factors such as comorbidities and anesthetic risks are assessed to ensure a favorable safety profile.

While this model remains theoretical, its future development could significantly transform the field of psychiatry, offering much more personalized and effective diagnoses and treatments based on each patient’s individual biology rather than solely on general clinical criteria.

The integration of biomarkers in DBS highlights the critical need for interdisciplinary collaboration in the field of psychiatry. As noted, combining expertise from neuroimaging, genetics, and clinical data is essential to develop personalized treatment protocols that address the unique biological and psychological profiles of patients. This collaborative approach not only enriches our understanding of psychiatric disorders but also enhances the precision of interventions.

For instance, the relationship between genetic variations, such as those in the serotonin transporter gene (5-HTTLPR), and their impact on neurotransmitter systems underscores the importance of genetic factors in the manifestation of psychiatric symptoms. The link between these genetic differences and neuroimaging findings, like altered amygdala activity, illustrates how biomarker research can provide a comprehensive view of a patient’s condition [11].

By recognizing these connections, researchers can better tailor treatments to the individual, potentially leading to more effective and targeted therapies. Such collaboration could pave the way for innovative strategies in precision psychiatry, ultimately improving patient outcomes. Highlighting these collaborative efforts is vital for advancing the field and ensuring that treatment protocols evolve to reflect the complex interplay of genetic, neurobiological, and clinical factors. 

## 5. Limitations

One of the most important limitations for the use of biomarkers in DBS is that psychiatric diagnoses often rely on subjective clinical criteria that overlap between disorders, such as in the DSM. This makes it difficult to pinpoint specific biomarkers, as the same symptoms can manifest in different conditions. The reliance on biomarkers for precision psychiatry may not overcome this inherent complexity, and it might still lead to misdiagnosis or incorrect treatment pathways. Also, while biomarkers show great potential, there is currently no standardized or generalized set of biomarkers for psychiatric illnesses, especially for use in DBS. Without consistent, reproducible biomarkers, the application of DBS remains theoretical. The lack of standardization may slow down the adoption of precision psychiatry approaches and limit their widespread clinical utility. In addition to this, integrating various biomarker data (e.g., genomic, proteomic, neuroimaging) into a meaningful “biosignature” for each patient poses significant technical challenges. This requires advanced computational tools for big data analysis and artificial intelligence, which may not yet be fully developed or accessible in all clinical settings. Additionally, the large volume of data could lead to errors or misinterpretations, potentially affecting treatment outcomes. The proposed workflow of precision psychiatry and DBS is theoretical and may face challenges in being generalized to the broader population since patients with psychiatric disorders are highly diverse, and the biosignatures that work for one patient may not apply to others. Finally, implementing such a comprehensive precision psychiatry model, involving advanced omic studies, neuroimaging, and DBS, would likely be expensive. The high cost could limit accessibility to these treatments, particularly in underfunded healthcare systems. This might create inequalities in who can receive personalized care, with DBS and precision psychiatry being available only to a small subset of patients who can afford these cutting-edge technologies. Finally, even with precision psychiatry, there are risks of complications, and predicting which patients may develop side effects or adverse reactions could be difficult. Additionally, DBS is a neurosurgical procedure that involves direct manipulation of brain activity, raising ethical concerns about autonomy, consent, and long-term impacts on patients’ personality or cognition.

## 6. Conclusions

In conclusion, although numerous studies have been carried out to identify biomarkers in psychiatric diseases, there is currently no broad standardization or generalization in their use for deep brain stimulation in psychosurgery.

The growing knowledge of these biomarkers and their application could facilitate the advance towards precision psychiatry in the coming decades. In this context, identifying the biosignatures that best adapt to DBS treatment will be especially important.

It is crucial to advance the personalization of the diagnosis of psychiatric diseases, focusing on the identification of dysfunctional neural networks that contribute to the patient’s symptoms and selecting the most appropriate treatment based on this. Establishing a “brain symptomatology” will be essential to direct DBS towards the correct pathological network.

In addition to being vital for the diagnosis and objective assessment of disease severity, biomarkers can also be valuable predictors of prognosis, providing more realistic expectations to the medical team, the patient, and their relatives.

Future biomarkers in psychosurgery are also expected to play an important role in treatment monitoring and objective assessment of DBS response during follow-up. Closed-loop systems, which integrate disease biomarker recording with self-programming and stimulation, appear to be a promising technology for psychiatric pathologies due to the high clinical variability of these diseases and the need for continuous monitoring.

Although DBS in psychosurgery still has a long way to go, it offers great hope for patients with severe psychiatric diseases who do not respond to other treatments. In the future, joint analysis of clinical/behavioral, omic, neuroimaging, electrophysiological, and neurobiochemical biomarkers is expected to allow the development of more personalized, precise, and effective DBS treatments.

In terms of future research directions, large-scale, multicenter clinical trials should be conducted to validate the efficacy and safety of identified biomarkers and treatment protocols, ensuring diverse representation of patient populations.

Also, methodologies for integrating various types of biomarkers (genomic, proteomic, neuroimaging) into a cohesive biosignature that can predict treatment outcomes more accurately are necessary.

The formulation of comprehensive ethical guidelines addressing the implications of precision psychiatry and the use of invasive treatments, ensuring patient rights and safety are prioritized, should also be developed and studied.

Finally, the application of machine learning algorithms to analyze complex datasets from biomarker studies, enabling more precise predictions about treatment efficacy and patient outcomes, should be explored and developed.

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
