# Peer review of "Biomarkers: The Key to Enhancing Deep Brain Stimulation Treatment for Psychiatric Conditions"

_brainsci, 2024, doi:10.3390/brainsci14111065_

Round 1
Reviewer 1 Report
Comments and Suggestions for Authors
Review of the paper "BIOMARKERS: THE KEY TO ENHANCING DBS TREATMENT FOR PSYCHIATRIC CONDITIONS,"
This is a very interesting subject
I read the manuscript with great interest
The paper can be accepted after major revision:
No abbreviation in title
More detailed evaluation covering several important aspects: clarity, methodology, findings, and implications for future research.
I request more clarity in the inclusion/exclusion criteria
the search strategy have to be specific
the databases and search terms have to be more explain
This detail would make the findings more pertinent
Integration of Biomarkers: Elaborate on how the identified biomarkers could be integrated into clinical practice. Discuss potential pathways for implementation and any existing obstacles.
Clarification of Findings: more detailed analysis of findings and implications would strengthen the argument for the importance of biomarkers in DBS treatment.
Add a section on the limitations of the research
The conclusion could be enhanced by outlining specific unanswered questions or future research directions. This would provide a clear path for upcoming studies in the field.
Author Response
- No abbreviation in title
Response:
We have revised the title from "BIOMARKERS: THE KEY TO ENHANCING DBS TREATMENT FOR PSYCHIATRIC CONDITIONS" to "BIOMARKERS: THE KEY TO ENHANCING DEEP BRAIN STIMULATION TREATMENT FOR PSYCHIATRIC CONDITIONS."
- I request more clarity in the inclusion/exclusion criteria. The search strategy have to be specific. The databases and search terms have to be more explain.
Response: Thank you for your insightful suggestion. In accordance with the PRISMA guidelines, we have provided more detailed information on the inclusion and exclusion criteria, which can now be found in section 2.1 (Materials and Methods). Additionally, we have included the specific search terms and databases used in the study.
- Integration of Biomarkers: Elaborate on how the identified biomarkers could be integrated into clinical practice. Discuss potential pathways for implementation and any existing obstacles.
Clarification of Findings: more detailed analysis of findings and implications would strengthen the argument for the importance of biomarkers in DBS treatment.
Response: We have expanded the discussion section to include a more detailed explanation of how biomarkers can be integrated into clinical practice, outlining potential implementation pathways. Additionally, I have addressed some of the obstacles that may arise in this process.
- Add a section on the limitations of the research
- Response: Please find a new section titled LIMITATIONS just after the DISCUSSION section.
- The conclusion could be enhanced by outlining specific unanswered questions or future research directions. This would provide a clear path for upcoming studies in the field.
Response: Thank you for your suggestion. In response, a final paragraph addressing this topic has been thoughtfully added to the Conclusion section.
Reviewer 2 Report
Comments and Suggestions for Authors
This article offers a valuable overview of deep brain stimulation (DBS) in treating psychiatric disorders, with a particular emphasis on identifying biomarkers for more personalized approaches. However, several weaknesses and limitations can be identified that, if addressed, could enhance its clarity, depth, and overall impact.
- The article primarily summarizes existing literature without introducing novel perspectives or interpretations. While such a review is useful, it falls short of advancing current knowledge or critically analyzing the existing gaps in the field. A more thorough evaluation of the limitations of current DBS applications and biomarker research would significantly strengthen the manuscript.
- While the promise of biomarkers is acknowledged, the article does not sufficiently address the challenges associated with translating these biomarkers into clinical practice. Additionally, the discussion surrounding how DBS modulates neural circuits is somewhat underdeveloped. There is an expanding body of research exploring the mechanisms through which DBS exerts its effects, particularly via network modulation (Neumann, 2023, Trends in Neurosciences).
- The section discussing biomarkers across various psychiatric disorders, such as schizophrenia, depression, and anxiety, lacks depth. A more systematic and comprehensive review of these biomarkers could enhance this section significantly. For instance, in schizophrenia, substantial literature highlights disruptions in oscillatory profiles and connectivity patterns as critical factors (Uhlhaas & Singer, 2010, Nature Reviews Neuroscience; Tarasi et al., 2023, Schizophrenia Bulletin; Kirihara et al., 2012, Biological Psychiatry; Friston, 2016, Schizophrenia Research). These disturbances in brain network synchrony and connectivity suggest important targets for DBS interventions but are not explored in detail.
- Although biomarkers are a central theme, the article does not effectively connect these findings to real-world applications. Providing concrete examples of how clinicians might adapt DBS treatments based on biomarker information would enhance the article's relevance. Moreover, emphasizing the importance of multidisciplinary collaboration—between neuroscientists, clinicians, and bioinformaticians—would further enhance the clinical applicability of the review.
- The development and application of biomarkers in DBS necessitate collaboration across various disciplines. Highlighting the importance of these collaborations is crucial, particularly in integrating neuroimaging, genetics, and clinical data. This teamwork is essential for creating more personalized treatment protocols. Biomarkers often correlate with genetic factors that can lead to neurotransmitter imbalances, ultimately reflecting in the neuroimaging and clinical symptoms observed (O'Hare, 2023, International Journal of Molecular Sciences; Zhang et al., 2024, International Journal of Molecular Sciences). For example, variations in the serotonin transporter gene (5-HTTLPR) have been linked to increased risks of anxiety and depression (Fratelli et al., 2020, Genes). Such genetic differences can influence brain processing, as neuroimaging studies have shown altered amygdala activity (Pezawas et al., 2005, Nature Neuroscience).
- While precision psychiatry is presented as a promising future direction, the article does not provide concrete examples or pathways for how DBS can be integrated into this approach. It would be beneficial to explore how clinicians can transition to a more personalized model of care and address the practical challenges they might face in doing so.
- Lastly, the review would greatly benefit from a more critical perspective on DBS, emphasizing both the strengths and weaknesses of current research and biomarker development. This could include discussions on conflicting evidence, unresolved questions, and ethical considerations surrounding the use of DBS in psychiatry.
The English level is generally acceptable.
Author Response
- The article primarily summarizes existing literature without introducing novel perspectives or interpretations. While such a review is useful, it falls short of advancing current knowledge or critically analyzing the existing gaps in the field. A more thorough evaluation of the limitations of current DBS applications and biomarker research would significantly strengthen the manuscript.
Response: Thank you for your insightful feedback. In response, we have made several revisions throughout the manuscript, with particular attention to your suggestion. A new section titled "LIMITATIONS" has been added immediately following the Discussion section, where we analyze the limitations of current DBS applications and biomarker research.
- While the promise of biomarkers is acknowledged, the article does not sufficiently address the challenges associated with translating these biomarkers into clinical practice. Additionally, the discussion surrounding how DBS modulates neural circuits is somewhat underdeveloped. There is an expanding body of research exploring the mechanisms through which DBS exerts its effects, particularly via network modulation (Neumann, 2023, Trends in Neurosciences).
Response: Thank you for your valuable input. In response, we have included information in the newly added "LIMITATIONS" section regarding the challenges of translating biomarkers into clinical practice. Additionally, as per your suggestion, we have incorporated a brief description in the Introduction section (1.1. Deep Brain Stimulation (DBS) and Psychosurgery) on how DBS modulates neural circuits, using the reference you kindly provided.
- The section discussing biomarkers across various psychiatric disorders, such as schizophrenia, depression, and anxiety, lacks depth. A more systematic and comprehensive review of these biomarkers could enhance this section significantly. For instance, in schizophrenia, substantial literature highlights disruptions in oscillatory profiles and connectivity patterns as critical factors (Uhlhaas & Singer, 2010, Nature Reviews Neuroscience; Tarasi et al., 2023, Schizophrenia Bulletin; Kirihara et al., 2012, Biological Psychiatry; Friston, 2016, Schizophrenia Research). These disturbances in brain network synchrony and connectivity suggest important targets for DBS interventions but are not explored in detail.
Response: Thank you for your insightful suggestion. In response, we have revised the section discussing biomarkers across various psychiatric disorders, such as schizophrenia, depression, and anxiety, to provide a more systematic and comprehensive review. We have also incorporated the references you mentioned, particularly focusing on disruptions in oscillatory profiles and connectivity patterns in schizophrenia, as highlighted by Uhlhaas & Singer (2010), Tarasi et al. (2023), Kirihara et al. (2012), and Friston (2016). These additions have helped to further explore the potential targets for DBS interventions in these disorders
- Although biomarkers are a central theme, the article does not effectively connect these findings to real-world applications. Providing concrete examples of how clinicians might adapt DBS treatments based on biomarker information would enhance the article's relevance. Moreover, emphasizing the importance of multidisciplinary collaboration—between neuroscientists, clinicians, and bioinformaticians—would further enhance the clinical applicability of the review.
Response: We have included in the DISCUSSION section a potential workflow describing how clinicians might adapt DBS treatments based on biomarker information. Also we have emphasized the importance of multidisciplinary collaboration in the DISCUSSION section.
- The development and application of biomarkers in DBS necessitate collaboration across various disciplines. Highlighting the importance of these collaborations is crucial, particularly in integrating neuroimaging, genetics, and clinical data. This teamwork is essential for creating more personalized treatment protocols. Biomarkers often correlate with genetic factors that can lead to neurotransmitter imbalances, ultimately reflecting in the neuroimaging and clinical symptoms observed (O'Hare, 2023, International Journal of Molecular Sciences; Zhang et al., 2024, International Journal of Molecular Sciences). For example, variations in the serotonin transporter gene (5-HTTLPR) have been linked to increased risks of anxiety and depression (Fratelli et al., 2020, Genes). Such genetic differences can influence brain processing, as neuroimaging studies have shown altered amygdala activity (Pezawas et al., 2005, Nature Neuroscience).
Response: We have added a paragraph on this topic in the DISCUSSION section at the end of it. Also, new references have been added (70 and 71).
- While precision psychiatry is presented as a promising future direction, the article does not provide concrete examples or pathways for how DBS can be integrated into this approach. It would be beneficial to explore how clinicians can transition to a more personalized model of care and address the practical challenges they might face in doing so.
Response: Thank you for your valuable feedback. In response to your suggestion, I have included a discussion at the end of the DISCUSSION section that outlines how biomarkers can be integrated into clinical practice. This section explores the pathways through which clinicians can transition to a more personalized model of care, addressing the practical challenges they may encounter in the process.
- Lastly, the review would greatly benefit from a more critical perspective on DBS, emphasizing both the strengths and weaknesses of current research and biomarker development. This could include discussions on conflicting evidence, unresolved questions, and ethical considerations surrounding the use of DBS in psychiatry.
Response: A final word on this topic has been added to the CONCLUSSION section. Here we explain future pathways of research in biomarkers. Also a note on ethical considerations has been included in the LIMITATIONS section.
Reviewer 3 Report
Comments and Suggestions for Authors
The study titled “Biomarkers: The Key to Improving DBS Treatment for Psychiatric Conditions” provided a narrative review of the role of biomarkers in improving deep brain stimulation (DBS) treatments, specifically for psychiatric conditions such as depression, OCD, and other severe disorders, compared to conventional therapies. This could have been made clearer in the title itself, as it seemed a bit biased in exploring the topic that the group of authors themselves worked on and citing the group’s findings.
In the introduction to the paper, the authors provided an adequate introduction to DBS and its use in psychosurgery. The discussion of psychiatric illnesses being network-based rather than isolated disorders is a significant point. However, the paper could have benefited from a deeper exploration of the limitations of current diagnostic methods and the DSM-V. While mentioning the challenges of diagnosis, it elaborates on why current methods are insufficient to strengthen the argument for biomarkers. In precision psychiatry: The emphasis on precision psychiatry is well-justified, as it highlights the individualized treatment approach. The connection made between biomarkers and precision psychiatry is strong, although the article could explore more concrete examples of success in precision medicine from other fields (such as oncology) to create a comparative understanding. In biomarker classification: The classification of biomarkers into clinical/behavioral, peripheral, neuroimaging, electrophysiological, and neurobiochemical is well organized and covers a wide range of research. However, some details seem underdeveloped. For example, peripheral biomarkers are briefly discussed, but their potential is still vague. The article could strengthen this section by providing more detailed examples or research studies that show how these markers have been or can be applied in real-world clinical settings.
The use of neuroimaging and electrophysiological markers: The discussion of neuroimaging and electrophysiological markers is one of the most frequently presented descriptions. It effectively links biomarkers with identification of specific neural circuits for more effective DBS targeting. This section could be further expanded by addressing the technological limitations and costs associated with advanced imaging techniques.
Challenges and future perspectives: The paper rightly points out the current lack of standardization in biomarkers and the complexity of psychiatric disorders. It discusses the challenges in adopting closed-loop systems, which are crucial for tailoring DBS in real time based on biomarker feedback. However, a discussion of closed-loop systems could benefit from more practical examples of where such systems have shown promise, even in preliminary research.
The conclusion is forward-looking, emphasizing the need for more personalized, biomarker-driven psychiatry. However, the paper could increase its impact by offering more specific recommendations for future research or clinical applications. For example, suggesting ways to integrate biomarkers into current treatment protocols or elaborating on how to overcome current obstacles would make the conclusion stronger.
Points that really need to be improved: Despite being a narrative review, the article is superficial on such a current and well-documented topic. Given the vast literature available in the mentioned data sources (PubMed, Embase, Google Scholar), an analysis could have been much more in-depth.
The descriptions of certain types of biomarkers, such as peripheral biomarkers, are brief and underexplored.
The review lacks a critical assessment of the technological and financial barriers to applying biomarkers in clinical practice.
More specific examples of practical applications and challenges related to the integration of biomarkers in psychiatric treatment could have significantly strengthened the article.
In addition to the informality of presenting context by topics, which detracts from a narrative on a subject, the lack of definition of acronyms in its first citation. And there is a lack of a link between the contexts raised without a conclusion on the subject.
This article is a solid contribution to the literature on biomarkers in DBS, but could benefit from a more in-depth exploration of practical applications, challenges, and future challenges to make it more impactful.
Comments on the Quality of English LanguageAdequate
Author Response
The study titled “Biomarkers: The Key to Improving DBS Treatment for Psychiatric Conditions” provided a narrative review of the role of biomarkers in improving deep brain stimulation (DBS) treatments, specifically for psychiatric conditions such as depression, OCD, and other severe disorders, compared to conventional therapies. This could have been made clearer in the title itself, as it seemed a bit biased in exploring the topic that the group of authors themselves worked on and citing the group’s findings.
In the introduction to the paper, the authors provided an adequate introduction to DBS and its use in psychosurgery. The discussion of psychiatric illnesses being network-based rather than isolated disorders is a significant point. However, the paper could have benefited from a deeper exploration of the limitations of current diagnostic methods and the DSM-V. While mentioning the challenges of diagnosis, it elaborates on why current methods are insufficient to strengthen the argument for biomarkers. In precision psychiatry: The emphasis on precision psychiatry is well-justified, as it highlights the individualized treatment approach. The connection made between biomarkers and precision psychiatry is strong, although the article could explore more concrete examples of success in precision medicine from other fields (such as oncology) to create a comparative understanding. In biomarker classification: The classification of biomarkers into clinical/behavioral, peripheral, neuroimaging, electrophysiological, and neurobiochemical is well organized and covers a wide range of research. However, some details seem underdeveloped. For example, peripheral biomarkers are briefly discussed, but their potential is still vague. The article could strengthen this section by providing more detailed examples or research studies that show how these markers have been or can be applied in real-world clinical settings.
The use of neuroimaging and electrophysiological markers: The discussion of neuroimaging and electrophysiological markers is one of the most frequently presented descriptions. It effectively links biomarkers with identification of specific neural circuits for more effective DBS targeting. This section could be further expanded by addressing the technological limitations and costs associated with advanced imaging techniques.
Challenges and future perspectives: The paper rightly points out the current lack of standardization in biomarkers and the complexity of psychiatric disorders. It discusses the challenges in adopting closed-loop systems, which are crucial for tailoring DBS in real time based on biomarker feedback. However, a discussion of closed-loop systems could benefit from more practical examples of where such systems have shown promise, even in preliminary research.
The conclusion is forward-looking, emphasizing the need for more personalized, biomarker-driven psychiatry. However, the paper could increase its impact by offering more specific recommendations for future research or clinical applications. For example, suggesting ways to integrate biomarkers into current treatment protocols or elaborating on how to overcome current obstacles would make the conclusion stronger.
Points that really need to be improved: Despite being a narrative review, the article is superficial on such a current and well-documented topic. Given the vast literature available in the mentioned data sources (PubMed, Embase, Google Scholar), an analysis could have been much more in-depth.
The descriptions of certain types of biomarkers, such as peripheral biomarkers, are brief and underexplored.
The review lacks a critical assessment of the technological and financial barriers to applying biomarkers in clinical practice.
More specific examples of practical applications and challenges related to the integration of biomarkers in psychiatric treatment could have significantly strengthened the article.
In addition to the informality of presenting context by topics, which detracts from a narrative on a subject, the lack of definition of acronyms in its first citation. And there is a lack of a link between the contexts raised without a conclusion on the subject.
This article is a solid contribution to the literature on biomarkers in DBS, but could benefit from a more in-depth exploration of practical applications, challenges, and future challenges to make it more impactful.
Response:
Thank you for your thoughtful and detailed feedback on our manuscript. We have carefully considered your suggestions and have made several important revisions to improve the quality and depth of the article. Specifically, we have included a section addressing the limitations of both biomarkers and imaging techniques, with a particular focus on the associated costs, as suggested. Additionally, we expanded the sections on the results you highlighted, providing further detail on areas we consider to be of significant importance.
In the discussion, we have also added a portion that explains how biomarkers can be applied in clinical practice, offering more concrete examples to strengthen the connection between theory and practical implementation. Furthermore, the conclusion has been revised to include a forward-looking perspective, where we provide specific recommendations for future research and clinical applications.
We greatly appreciate your constructive recommendations, which have significantly enhanced our manuscript. Thank you again for your valuable input.
Round 2
Reviewer 1 Report
Comments and Suggestions for Authors
Review of the paper " BIOMARKERS: THE KEY TO ENHANCING DEEP BRAIN STIMULATION TREATMENT FOR PSYCHIATRIC CONDITIONS" version 2
The paper can be accepted
I think is very interesting paper providing details of various biomarkers related to psychiatric conditions and deep brain stimulation with structured organization of different biomarkers
It aligns the current advancements in precision medicine.
This is a timely and relevant topic in neuroscience, especially in the treatment of severe and treatment-resistant psychiatric disorders.
The authors adhere to PRISMA guidelines, which adds credibility to their systematic approach. Their inclusion and exclusion criteria, and the thorough exploration of available literature, demonstrate a well-organized review process.
The article highlights future directions, especially the promise of closed-loop DBS systems and machine learning to personalize treatments based on biomarkers. This forward-looking perspective could spark new research in the field.
Author Response
The paper can be accepted
I think is very interesting paper providing details of various biomarkers related to psychiatric conditions and deep brain stimulation with structured organization of different biomarkers
It aligns the current advancements in precision medicine.
This is a timely and relevant topic in neuroscience, especially in the treatment of severe and treatment-resistant psychiatric disorders.
The authors adhere to PRISMA guidelines, which adds credibility to their systematic approach. Their inclusion and exclusion criteria, and the thorough exploration of available literature, demonstrate a well-organized review process.
The article highlights future directions, especially the promise of closed-loop DBS systems and machine learning to personalize treatments based on biomarkers. This forward-looking perspective could spark new research in the field.
RESPONSE: Dear Reviewer,
Thank you for your thoughtful feedback and kind words. We’re glad you found the paper interesting and aligned with current advancements in precision medicine and psychiatric treatments using DBS. Your recognition of our structured approach, and focus on future directions, such as closed-loop systems and machine learning, is greatly appreciated.
We are excited to see the paper accepted and grateful for your insights.
Reviewer 2 Report
Comments and Suggestions for Authors
The authors have done a great job addressing my concerns. The only inaccuracy is related to the added references. You mentioned that you had added two new references ("new references have been added (70 and 71)"). However, they pertain to the first version of the manuscript (unless you simply forgot to highlight them). Additionally, in the section on schizophrenia, some references that you indicated in the rebuttal as being integrated are missing.
Author Response
The authors have done a great job addressing my concerns. The only inaccuracy is related to the added references. You mentioned that you had added two new references ("new references have been added (70 and 71)"). However, they pertain to the first version of the manuscript (unless you simply forgot to highlight them). Additionally, in the section on schizophrenia, some references that you indicated in the rebuttal as being integrated are missing.
RESPONSE: Dear Reviewer,
Thank you for your time, valuable feedback, and for pointing out the inaccuracy regarding the references. We apologize for the confusion and have now corrected the issue. The new references mentioned (70 and 71) in our original response actually correspond to references 11 and 78, which are now correctly highlighted in the reference section.
Additionally, we have ensured that the new references, including those related to schizophrenia (57, 58, 59, 60), are highlighted and properly integrated throughout the manuscript.
We appreciate your thorough review and hope these corrections address your concerns.
Reviewer 3 Report
Comments and Suggestions for Authors
The authors stated as the objective of the study a review of the existing literature on related to deep brain stimulation in the treatment of psychiatric disorders as well as other biomarkers already described in these diseases, which could have a possible future application in psychosurgery using this therapeutic method, but this was far from what was shown. This article is not a systematic review, it did not follow the PRISMA guideline for systematic reviews as they state in the method and it does not have any scientific rigor of the systematic review, neither in the search nor in the analysis. The most they can do is transform it into a narrative review, but to say that it is systematic in the method is completely wrong and this needs to be changed, because with these databases that they searched, certainly three times as many articles would be found and the result would be much more robust than what was presented. The result is a superficial analysis of the theme shown primarily by topics and not by narrative of the subject, showing a little more robustness of content and narrative of the context and current findings of a review, it does not highlight what has changed over time due to superficiality and lack of exploration of the literature as proposed in the method. Some result topics such as 3.1 and 3.5, the authors cite only 4 studies, this shows how much has not been explored in the literature on the subject. In my opinion, the authors got lost in the subject, I had previously suggested that instead of doing a review of several psychiatric conditions, they should do one or two in particular, elaborating in a deep and comprehensive way and not in the way that was done of everything and confusing, without a comprehensive conclusion.
Comments on the Quality of English LanguageAdequate
Author Response
The authors stated as the objective of the study a review of the existing literature on related to deep brain stimulation in the treatment of psychiatric disorders as well as other biomarkers already described in these diseases, which could have a possible future application in psychosurgery using this therapeutic method, but this was far from what was shown. This article is not a systematic review, it did not follow the PRISMA guideline for systematic reviews as they state in the method and it does not have any scientific rigor of the systematic review, neither in the search nor in the analysis. The most they can do is transform it into a narrative review, but to say that it is systematic in the method is completely wrong and this needs to be changed, because with these databases that they searched, certainly three times as many articles would be found and the result would be much more robust than what was presented. The result is a superficial analysis of the theme shown primarily by topics and not by narrative of the subject, showing a little more robustness of content and narrative of the context and current findings of a review, it does not highlight what has changed over time due to superficiality and lack of exploration of the literature as proposed in the method. Some result topics such as 3.1 and 3.5, the authors cite only 4 studies, this shows how much has not been explored in the literature on the subject. In my opinion, the authors got lost in the subject, I had previously suggested that instead of doing a review of several psychiatric conditions, they should do one or two in particular, elaborating in a deep and comprehensive way and not in the way that was done of everything and confusing, without a comprehensive conclusion.
RESPONSE: Dear reviewer
We greatly appreciate your comments and the time you have dedicated to reviewing our work. Our main objective is to provide a general overview that highlights the importance of biomarkers in any invasive neurosurgical intervention for the treatment of psychiatric disorders. The advancement of surgery in this field faces numerous challenges, including the subjectivity of clinical scales, the possibility that diseases may be influenced by the placebo effect, and the fundamental need for biomarkers.
We do not intend for our presentation to be exhaustive, as this would go beyond the scope of the objective. However, we have adhered to the recommended search guidelines to maintain rigor in our approach.
Thank you once again for your valuable suggestions, which undoubtedly contribute to improving the quality of the work.